# Dual role of the S5 segment in type 1 ryanodine receptor channel gating
Takashi Murayama [1] ✉, Yuya Otori[2], Nagomi Kurebayashi [1], Toshiko Yamazawa [3], Hideto Oyamada[4], Takashi Sakurai [1] & Haruo Ogawa [2] ✉

The type 1 ryanodine receptor (RyR1) is a $Ca^{2+}$ release channel in the sarcoplasmic reticulum that is essential for skeletal muscle contraction. RyR1 forms a channel with six transmembrane segments, in which S5 is the fifth segment and is thought to contribute to pore formation. However, its role in channel gating remains unclear. Here, we performed a functional analysis of several disease-associated mutations in S5 and interpreted the results with respect to the published RyR1 structures to identify potential interactions associated with the mutant phenotypes. We demonstrate that S5 plays a dual role in channel gating: the cytoplasmic side interacts with S6 to reduce the channel activity, whereas the luminal side forms a rigid structural base necessary for S6 displacement in channel opening. These results deepen our understanding of the molecular mechanisms of RyR1 channel gating and provide insight into the divergent disease phenotypes caused by mutations in S5.

The type 1 ryanodine receptor (RyR1) is a $Ca^{2+}$ release channel in the sarcoplasmic reticulum (SR) of skeletal muscles and plays a central role in muscle contraction[1,2]. During excitation-contraction (E-C) coupling, the depolarization of the T-tubule membrane opens the RyR1 channel to release $Ca^{2+}$ via physical interactions with the α1 subunit (Cav1.1) of the dihydropyridine receptor (DHPR)[3,4]. This process is known as depolarization-induced $Ca^{2+}$ release (DICR). The RyR1 channel can also be directly activated by $Ca^{2+}$ binding, a phenomenon referred to as $Ca^{2+}$-induced $Ca^{2+}$ release (CICR)[5,6]. Mutations in the *RYR1* gene are associated with severe muscle diseases including malignant hyperthermia (MH)[7] and central core disease (CCD)[8]. Mutations associated with MH generally cause gain of function of the channel, in which inhalational anesthetics trigger massive $Ca^{2+}$ release to cause muscle contracture and high fever[7]. In most CCD cases, the loss of E-C coupling leads to the disease phenotype of muscle weakness and myopathy[9]. Therefore, MH and CCD mutations have opposing functional effects on channel activity.

RyR is an extensive (~2 MDa) homotetrameric protein complex featuring a substantial N-terminal cytoplasmic structure and six transmembrane segments (S1-S6) at its carboxyl (C) terminus, which form a cation channel[2,10]. Among these segments, S6 constitutes both a gate and a pore, whereas S1-S4 regulate the gate through the S4-S5 linker. Recent cryo-electron microscopy (EM) structures of RyR resolved at near-atomic resolution have revealed a complex architectural arrangement[11–13] and provided insights into the conformational changes associated with channel opening

by $Ca^{2+}$ [14,15]. By integrating high-resolution cryo-EM structures with quantitative functional analyses of mutant channels, we recently identified several important interactions in channel gating that span from the cytosolic domains to the transmembrane domains[16–18]. Thus, the combinatorial approach proved to be a robust strategy for unraveling the gating mechanism of RyR channels.

S5 is the fifth transmembrane helix connecting the S4-S5 linker to S6[2,10]. Although S5 is considered important for pore formation, its specific role in channel gating remains unknown. S5 is a major locus for disease-associated mutations, with over ten mutations reported in S5 and the subsequent beginning of the S5-S6 loop of RyR1 (Supplementary Table 1). Interestingly, some mutations are associated with MH, whereas others are linked with CCD. This raises the possibility that S5 may have different effects on channel function. To test this hypothesis, we conducted functional analyses of channels carrying disease-associated mutations using a heterologous expression system in HEK293 cells. We assessed the CICR activity of the mutant channels using intracellular $Ca^{2+}$ measurements and [³H]ryanodine binding assays[19,20]. Additionally, DICR activity was determined using a recently developed reconstituted platform[21]. Subsequently, the underlying mechanisms were interpreted based on the near-atomic structure of RyR1 and validated using mutant analysis. Our results suggest that S5 may play a dual role in the gating of the RyR1 channel; the cytoplasmic side interacts S6 to reduce the channel activity, whereas the luminal side forms a rigid structural base necessary for the displacement of S6 in the channel opening.

[1]Department of Cellular and Molecular Pharmacology, Juntendo University Graduate School of Medicine, Tokyo, 113-8421, Japan. [2]Department of Structural Biology, Graduate School of Pharmaceutical Sciences, Kyoto University, Kyoto, 606-8501, Japan. [3]Core Research Facilities, The Jikei University School of Medicine, Tokyo, 105-8461, Japan. [4]Pharmacological Research Center, Showa University, Tokyo, 142-8555, Japan. ✉e-mail: takashim@juntendo.ac.jp; haru@pharm.kyoto-u.ac.jp

**Fig. 1 | Location of disease-associated mutations on the RyR1 structure. a** Structure of RyR1 in the closed state (PDB accession code, 5TB0) viewed from the direction parallel to the lipid bilayer. **b** Magnified view of the dotted box in (**a**). Mutated residues for MH (orange) and CCD (light blue) are depicted in red spheres.

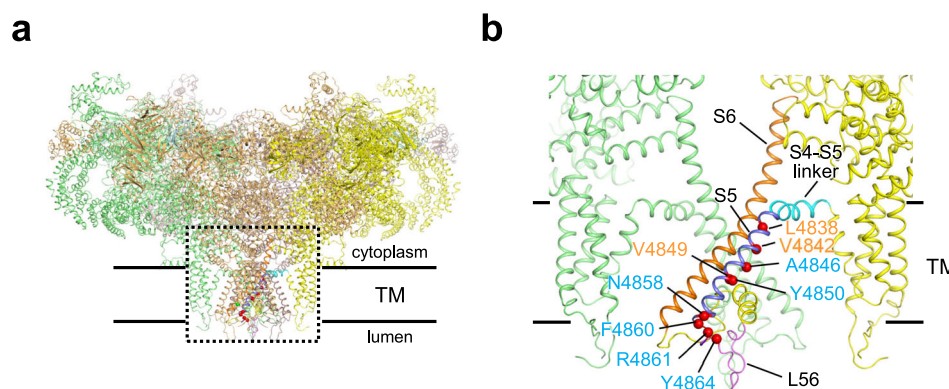

## Results

In our search of the mutation database (https://www.cardiodb.org/paralogue_annotation/), we identified 13 reported disease-associated mutations in S5 and the subsequent beginning of the S5-S6 luminal loop in the *RYR1* gene (Supplementary Table 1). Among these, we selected 11 missense mutations at nine different positions (Fig. 1a, b and Supplementary Table 1). Two deletion mutations (F4860del [22] and 4863-4869delYins [23]) were excluded because of complexities in considering the structural changes caused by mutations. Notably, 3 mutations (L4838V [24], V4842M [25] and V4849I [26]) were associated with MH, whereas the other 8 mutations (A4846V [27], F4850C [28], N4858D [29], F4860V [30], R4861C [22,31], R4861H [22,32], Y4864C [33] and Y4864H [34]) were associated with CCD (Fig. 1b and Supplementary Table 1). Mutant channels were created with rabbit RyR1 and stably expressed in HEK293 cells using a tetracycline-inducible expression system [19,20]. Western blot analysis confirmed expression of the mutant RyR1s with gel mobility similar to that of WT (Supplementary Fig. 1a). Because this region is well conserved between human and rabbit, the residue numbers of the mutations follow those of human RyR1 (Supplementary Table 1).

### Cellular $Ca^{2+}$ homeostasis of the mutant RyR1 channels

The phenotypes of mutant RyR1 channels were initially tested by caffeine-induced $Ca^{2+}$ release, a useful measure of CICR activity (Fig. 2a). Gain-of-function mutants generally exhibit an increased sensitivity to caffeine, whereas loss-of-function mutants show no or reduced response [35,36]. Typical fluo-4 $Ca^{2+}$ signals are shown in Fig. 2b. Caffeine triggered $Ca^{2+}$ transients in a dose-dependent manner in WT cells. The MH mutant V4842M showed an increased sensitivity to caffeine with a reduced peak. The CCD mutant R4861C exhibited reduced sensitivity to caffeine, with a similar peak value to WT. Another CCD mutant, Y4850C, exhibited virtually no $Ca^{2+}$ signals, even with 30 mM caffeine. We plotted the peak $Ca^{2+}$ signals by 10 mM caffeine. They were significantly reduced in three MH mutants (L4838V, V4842M and V4849I) and two CCD mutants (N4858D and Y4864C) (Fig. 2c). No $Ca^{2+}$ signals were detected in any of the three CCD mutants (Y4850C, F4860V and Y4864H). The averaged $Ca^{2+}$ signals from all the pooled data were plotted against each caffeine dose to calculate $EC_{50}$ values (Fig. 2d). $EC_{50}$ value was decreased in the three MH mutants and a CCD mutant (A4846V) and increased in two CCD mutants (R4861C and R4861H) (Fig. 2e).

Gain-of-function RyR1 mutants expressed in HEK293 cells cause $Ca^{2+}$ leak to reduce ER $Ca^{2+}$ ($[Ca^{2+}]_{ER}$) [19,20]. We therefore measured resting $[Ca^{2+}]_{ER}$ using R-CEPIA1er, a genetically-encoded ER $Ca^{2+}$ indicator [37]. $[Ca^{2+}]_{ER}$ was reduced to varying degrees in the three MH mutants (Fig. 2f). This is consistent with the reduced peak $Ca^{2+}$ signals by caffeine in these mutants (see Fig. 2c). In contrast, the CCD mutants did not substantially alter $[Ca^{2+}]_{ER}$. Reduction in $[Ca^{2+}]_{ER}$ activates store-operated $Ca^{2+}$ entry to increase resting cytoplasmic $Ca^{2+}$ ($[Ca^{2+}]_{cyt}$) [38]. The resting $[Ca^{2+}]_{cyt}$ was measured using fura-2, a ratiometric fluorescent $Ca^{2+}$ indicator. $[Ca^{2+}]_{cyt}$ was substantially higher in the three MH mutants than in WT, but not in the CCD mutants (Fig. 2g).

### [$^3$H]Ryanodine binding of the mutant RyR1 channels

Next, we examined [$^3$H]ryanodine binding, which reflects the $Ca^{2+}$-dependent channel activity [19,20]. WT RyR1 exhibited biphasic $Ca^{2+}$ dependence, with a peak value of approximately 0.05 (Fig. 3a). Notably, the three MH mutants exhibited an enhanced binding compared to WT, indicating a gain-of-function phenotype (Fig. 3a). Among the CCD mutants, A4846V showed reduced binding, whereas Y4850C exhibited no binding (Fig. 3b). The peak binding at pCa 4.5 was increased with two MH mutants (V4842M and V4849I) but reduced with A4846V (Fig. 3c). No measurable binding was observed with six CCD mutants (F4850C, N4858D, F4860V, R4861C, Y4864C and Y4864H), indicating that they are a severe loss-of-function phenotype (Fig. 3c). MH mutants at N-terminal and central regions show enhanced $Ca^{2+}$ sensitivity for activation and reduced $Ca^{2+}$ sensitivity for inactivation [19,20]. $EC_{50}$ was reduced with two MH mutants (L4838V and V4849I) (Fig. 3d), whereas $IC_{50}$ was increased with the three MH mutants (Fig. 3e). Based on these parameters, the activity of the mutant channels at resting $[Ca^{2+}]_{cyt}$ (pCa 7) was calculated and expressed as a relative value to that of WT [19,20]. The three MH mutants, L4838V, V4842M and V4849I, showed 10-, 2-, and 10-fold higher activity than WT, respectively (Fig. 3f), suggesting that they are mild (V4842M) and severe (L4838V and V4849I) gain-of-function mutations.

### DICR activity of the mutant RyR1 channels

Loss of DICR is a major cause for the CCD disease phenotypes [9,39]. Therefore, we measured DICR activity using a recently developed reconstituted platform [21]. HEK293 cells stably expressing RyR1 were infected with baculovirus for essential components (Cav1.1, β1a, Stac3 and JP2), Kir2.1 and R-CEPIA1er, and the activity was evaluated using the high potassium ($[K^+]$)-induced $Ca^{2+}$ release while monitoring $[Ca^{2+}]_{ER}$ (Fig. 4a). WT cells showed a $[K^+]$-dependent reduction in the fluorescence intensity of R-CEPIA1er (Fig. 4b). We plotted the $[K^+]$ dependence of the fluorescence changes corrected for the initial $[Ca^{2+}]_{ER}$ (Fig. 4c, d). This provided $EC_{50}$ for $[K^+]$ (Fig. 4e). The MH mutant V4849I exhibited reduced initial fluorescence (Fig. 4b, c) and $EC_{50}$ (Fig. 4e). A smaller $EC_{50}$ value was observed for the other MH mutants, L4838V and V4842M (Fig. 4c, e). In contrast, the CCD mutant R4861C exhibited increased $EC_{50}$ (Fig. 4d, e). Another CCD mutant, Y4850C, showed no reduction in $[Ca^{2+}]_{ER}$ by $[K^+]$ (Fig. 4b, d). No reduction in $[Ca^{2+}]_{ER}$ was observed in any of the four CCD mutants (N4858D, F4860V, Y4864C and Y4864H) (Fig. 4e). Taken together, these findings suggest that DICR activity is enhanced in MH mutants but reduced or lost in CCD mutants. The severity of the DICR phenotype in the CCD mutants was consistent with that of the CICR phenotype; Y4850C, N4858D, F4860V, Y4864C and Y4864H were more severe than A4846V or R4861C. R4861H showed no change in DICR activity.

The results of the functional assays for each mutant are summarized in Supplementary Table 3. The three MH mutants (L4838V, V4842M and V4849I) consistently showed increased caffeine sensitivity, reduced $[Ca^{2+}]_{ER}$, elevated $[Ca^{2+}]_{cyt}$, enhanced [$^3$H]ryanodine binding and enhanced DICR, clearly indicating a gain-of-function phenotype. In the CCD mutants, A4846V showed slightly reduced [$^3$H]ryanodine binding but

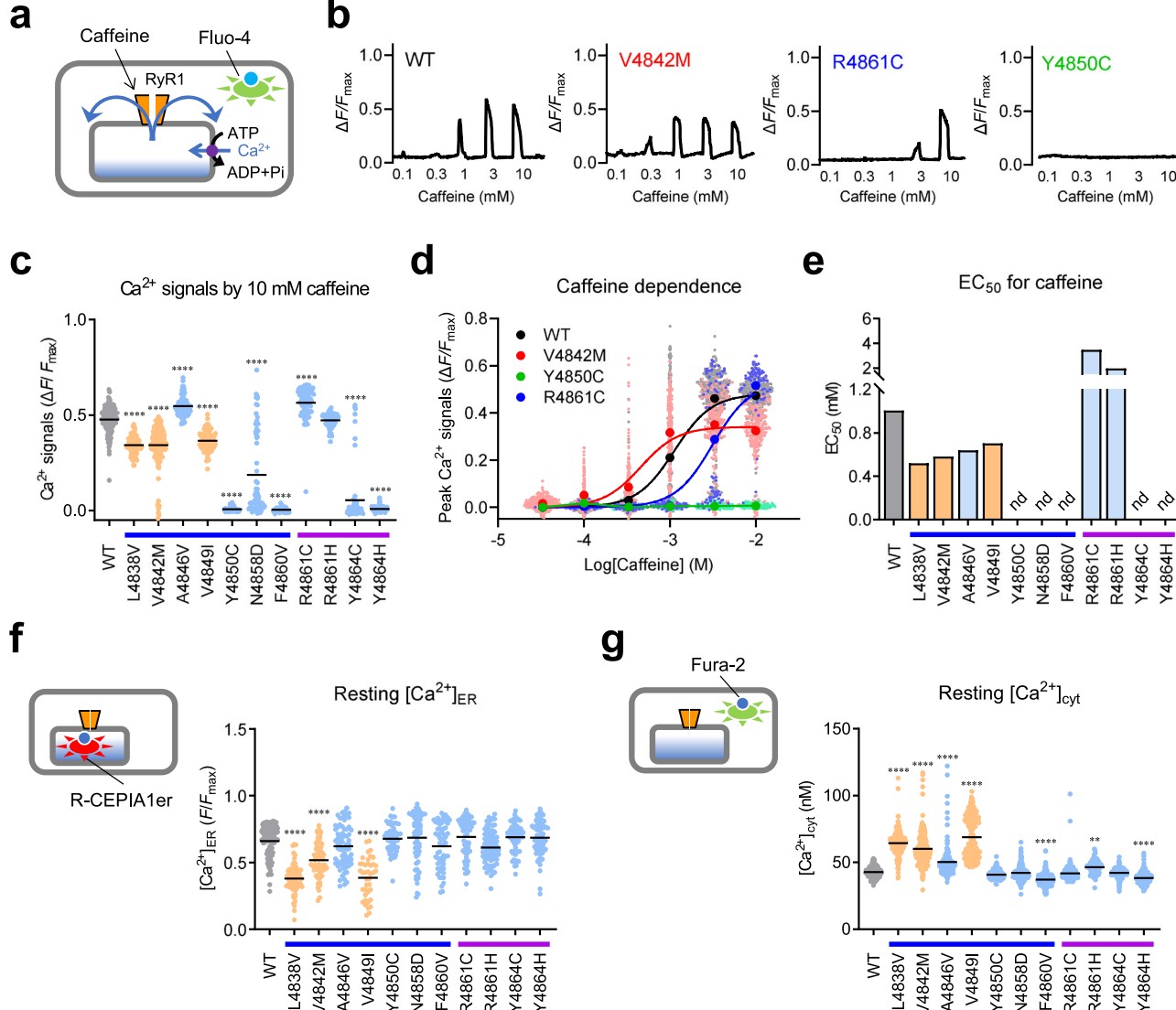

**Fig. 2 | Intracellular $Ca^{2+}$ measurements in cells expressing mutant RyR1 channels. a** Schematic drawing of caffeine-induced $Ca^{2+}$ release. HEK293 cells expressing the WT and mutant RyR1 channels were loaded with fluo-4 AM, and the $Ca^{2+}$ release via RyR1 was elicited by different concentrations (0.1–10 mM) of caffeine. **b** Representative traces of fluo-4 signals for WT and three mutants (V4842M, R4861C and Y4850C). The numbers under the trace indicate the concentrations of caffeine. **c** The peak $Ca^{2+}$ signals by 10 mM caffeine for WT (grey), MH (orange: L4838V, V4842M and V4849I) and CCD (light blue: A4846V, Y4850C, N4858D, F4860V, R4861C, R4861H, Y4864C and Y4864H) mutant RyR1 cells. **d** Caffeine-dependent $Ca^{2+}$ response in WT and three mutants cells (V4842M, Y4850C and R4861C). The averaged peak $Ca^{2+}$ signals were plotted against the caffeine concentrations and fitted to the dose-response curve. **e** $EC_{50}$s for caffeine in WT and mutant RyR1 cells. nd, not determined due to virtually no $Ca^{2+}$ signals by caffeine.

**f** Resting $[Ca^{2+}]_{ER}$ measurement using R-CEPIA1er. MH mutants showed lower $[Ca^{2+}]_{ER}$ than WT. **g** Resting $[Ca^{2+}]_{cyt}$ measurement using fura-2. MH mutants showed higher $[Ca^{2+}]_{cyt}$ than WT. For panels (**c, f, g**), data are shown as the means with individual points and were analyzed by one-way ANOVA with Dunnett's multiple comparisons test. $^{**}p < 0.01$, $^{****}p < 0.0001$ compared with WT. The number of cells (n), the number of independent dishes (N) and exact $p$-values in each experiment are summarized in Supplementary Table 2. For panel (**d**), data are shown as means and individual points (WT: $n = 230$, $N = 4$; V4842A: $n = 210$, $N = 4$; Y4850C: $n = 70$, $N = 3$; R4861C: $n = 70$, $N = 3$). For panel (**e**), $EC_{50}$s were calculated using individual averaged values for each caffeine concentration from all pooled data as shown in **d**. Blue and purple lines below the horizontal axis in (**c, e–g**) represent S5 and the S5-S6 luminal loop, respectively.

unchanged DICR. R4861C exhibited reduced caffeine sensitivity, no [$^3$H] ryanodine binding and reduced DICR, whereas R4861H exhibited reduced caffeine sensitivity without substantial changes in the other parameters. These mutations appear to result in mild loss-of-function phenotypes of varying degrees. In contrast, the remaining five CCD mutants (Y4850C, N4858D, F4860V, Y4864C and Y4864H) did not exhibit any measurable caffeine-induced $Ca^{2+}$ release, [$^3$H]ryanodine binding, or DICR, suggesting a severe loss-of-function phenotype.

### Molecular mechanisms of alterations by the mutations

To address the molecular mechanisms underlying the changes caused by the mutations, we used the published high-resolution structures of RyR1 in the

closed (PDB accession code, 5TB0) and open (PDB accession code, 5T15) states obtained by cryo-EM[14]. When the channel was opened, the cytoplasmic side of S6 and the S4-S5 linker move outward to open the gate (Fig. 5a and Supplementary Movie 1). The cytoplasmic side of S5 also moves outward. In contrast, the luminal side of S5 shows no movement upon channel opening. Three MH mutations (L4838V, V4842M and V4849I) were localized to the cytoplasmic side of S5 and were aligned linearly from the cytoplasm side into the membrane (Fig. 5b and Supplementary Movie 1). In the closed state, we identified the possible van der Waals interactions (L4838-I4928, V4842-I4933 and V4849-V4892) between S5 and the pore helix or S6 in the closed state and the open state (Fig. 5c, Supplementary Fig. 2, and Supplementary Movie 2). L4838V, V4842M and

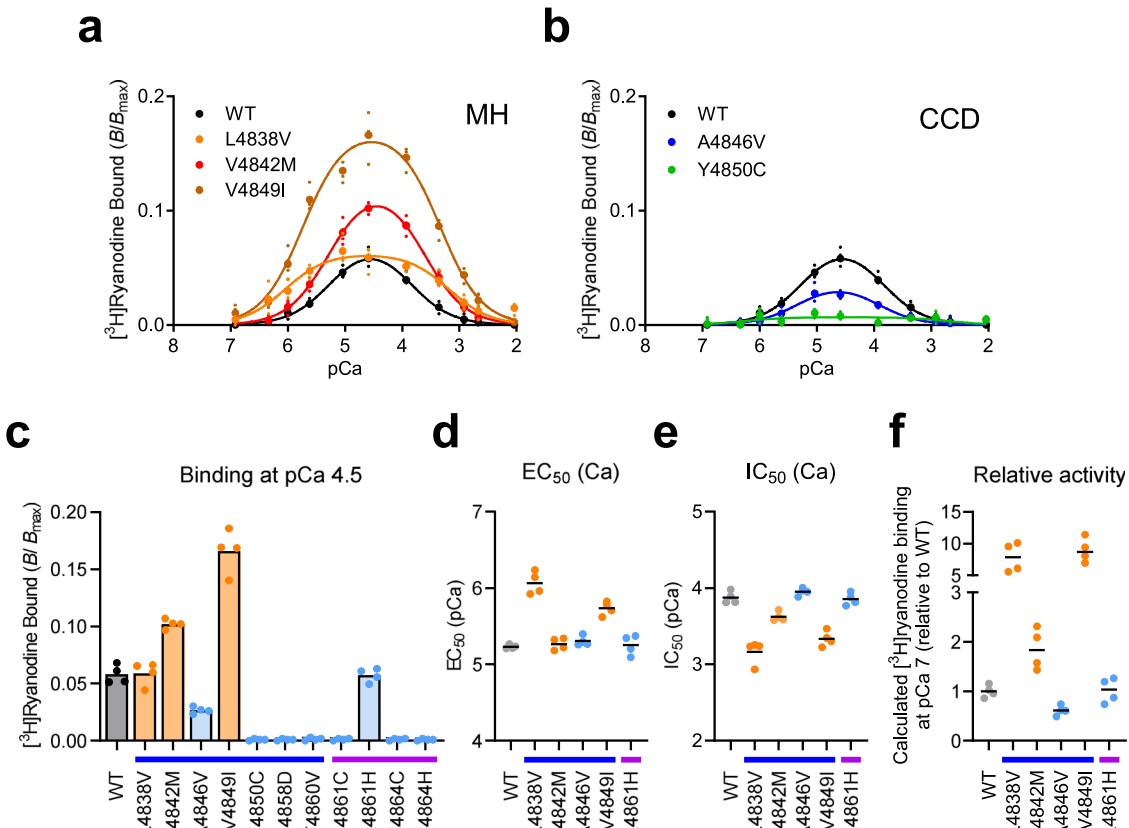

**Fig. 3 | Ca²⁺-dependent [³H]ryanodine binding of mutant RyR1 channels.**
**a**, **b** Microsomes from HEK293 cells expressing WT and mutant RyR1 (L4838V, V4842M and V4849I in **a** and A4846V and Y4850C in **b**) were incubated with 5 nM [³H]ryanodine for 5 h at 25 °C in the reaction medium at various concentrations of free Ca²⁺. **c** The [³H]ryanodine binding of microsomes from HEK293 cells expressing WT (black), MH (orange: L4838V, V4842M and V4849I) and CCD (light blue: A4846V, Y4850C, N4858D, F4860V, R4861C, R4861H, Y4864C and Y4864H)

mutant RyR1 at pCa 4.5. Note that the MH mutants show greater [³H]ryanodine binding than WT, whereas CCD mutants exhibited no or reduced binding. **d**, **e** EC₅₀ and IC₅₀ values for Ca²⁺. **f** Calculated [³H]ryanodine binding at resting [Ca²⁺]cyt (pCa 7) of mutant RyR1s representing relative activity to WT. Data are shown as the means and individual points (n = 4, N = 2). "n" is the number of assays, and "N" is the number of independent experiments. Blue and purple lines below the horizontal axis in (**c–f**) represent S5 and the S5-S6 luminal loop, respectively.

V4849I mutations may alter these interactions. Because these three mutants caused gain of function, the interactions formed by these amino acid residues may either stabilize the channel in the closed state or destabilize it in the open state, or both. In the luminal side of S5, we identified multiple interactions with the adjacent transmembrane segments of their own and neighboring subunits: possible hydrogen bonds (Y4850-N4575, R4861-Y4630 and Y4864-H4887) with S1', the S1-S2 loop and S4 of the neighboring subunit, cation-π interaction (N4858-F4808) with S4 of the neighboring subunit and van der Waals interaction (F4860-F4917) with S6 (Fig. 5d and Supplementary Movie 2). Because all these interactions were also preserved in the open state (Supplementary Fig. 2), these interactions may contribute to the formation of a rigid structural base, which is necessary for the displacement of the cytoplasmic side of S6 in channel opening. All disease-associated mutations may disrupt these interactions, leading to loss-of-function phenotypes.

**Validation of the molecular mechanism by mutants in the interacting partners**
To validate the above hypothesis, we conducted functional assays on RyR1 channels carrying mutations in their potential interacting partners. Notably, a search of the mutation database identified three myopathy-related mutations (H4887Y[40], F4808L[41], F4917L[42] and I4928V[25]) in the interacting partners (Supplementary Table 4). All the three mutations were expected to disrupt these interactions. We generated these disease-associated mutations, except for I4928V. In the remaining interacting partners, alanine-substituted mutations (Y4630A, I4928A and I4933A) were created. For V4892, isoleucine-substituted mutant (V4892I) was generated since the

mutant in the interaction partner (V4849I) is an isoleucine substitute. The expression of the mutant channels was confirmed by Western blotting (Supplementary Fig. 1b) and the channel activity was evaluated by the DICR assay. Resting [Ca²⁺]ER levels were substantially reduced in I4933A cells (Fig. 6a). A [K⁺]-dependent plot of [Ca²⁺]ER changes revealed that V4892A, I4928A and I4933A, mutants at the interacting partners for MH mutations, were more sensitive to [K⁺] than the WT, an index of the gain-of-function phenotype (Fig. 6b–e). Mutants at interacting partners associated with CCD mutations exhibited loss-of-function phenotypes to varying degrees: [K⁺]-induced Ca²⁺ release was reduced (F4808L) or lost (H4887Y) and the EC₅₀ for [K⁺] was increased (F4808L and F4917L) (Fig. 6b–f). An exception was Y4630A, which showed no difference in activity compared to WT.

We also assessed the caffeine sensitivity of the mutant channel in the reconstituted platform as an index of CICR activity (Supplementary Fig. 3). Caffeine sensitivity was enhanced in mutants with interacting partners associated with MH (Supplementary Fig. 3a) but was reduced or lost in those with interacting partners associated with CCD (Supplementary Fig. 3b, c). Taken together, the results of the mutant channels in the interacting partner correspond to those of the mutant channels in S5, strongly supporting our hypothesis that the proposed interactions are important for channel gating.

**Discussion**
In this study, we investigated the functional properties of a RyR1 channel carrying 11 disease-associated mutations (3 MH and 8 CCD mutations) in S5 using the HEK293 cell system. We demonstrated that MH mutations in the cytoplasmic side of S5 caused gain-of-function mutations, whereas CCD mutations in the luminal side exhibited a severe loss-of-function phenotype.

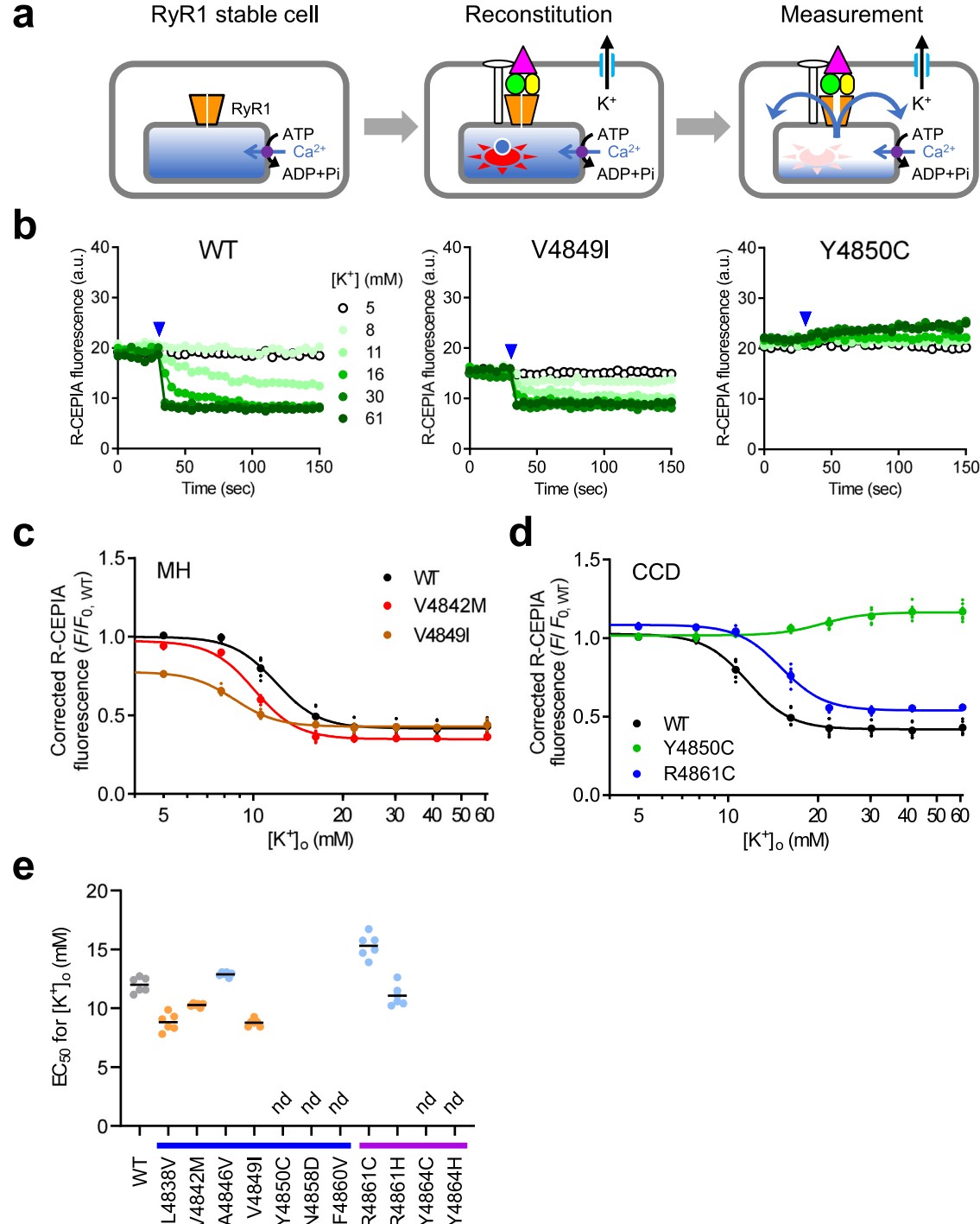

**Fig. 4 | DICR activity of the mutant RyR1 channels. a** Schematic drawing of the reconstitution and measurement of DICR. HEK293 cells expressing WT and mutant RyR1s were infected with baculovirus carrying essential components (Cav1.1, β1a, Stac3 and JP2) to reconstitute DICR machinery. R-CEPIA1er and Kir2.1 were also transduced to measure $[Ca^{2+}]_{ER}$ and hyperpolarize the membrane potential, respectively. DICR was triggered by depolarization of the membrane potential with a high $[K^+]$ solution. **b** Typical results of time-lapse R-CEPIA1er fluorescence measurement for WT (left), V4849I (center) and Y4850C (right) RyR1 cells. High $[K^+]$ solutions ranging from 5 to 61 mM (symbols shown in WT) were applied 30 s after starting (blue arrowheads). **c, d** $[K^+]$ dependence of R-CEPIA1er fluorescence corrected by $[Ca^{2+}]_{ER}$ in WT and MH (V4842M and V4849I) (**c**) and CCD (Y4850C and R4861C) (**d**) mutant RyR1 cells. **e** $EC_{50}$ values for $[K^+]$ of WT (black), MH (orange) and CCD (light blue) mutant RyR1s. nd, not determined due to virtually no reduction in $[Ca^{2+}]_{ER}$ by $[K^+]$. Data are shown as the means and individual points ($n = 6$, $N = 2$). "n" is the number of wells, and "N" is the number of independent experiments. Blue and purple lines below the horizontal axis represent S5 and the S5-S6 luminal loop, respectively.

Analysis of high-resolution RyR1 structures identified several possible interactions between residues responsible for disease-associated mutations and those in the adjacent transmembrane segments of their own or neighboring subunits. Our results suggest that S5 may play a dual role in the gating of the RyR1 channel: the cytoplasmic side reduces the channel activity, whereas the luminal side is essential for channel opening (Fig. 7).

We used multilateral approaches to evaluate the CICR activity of the disease-associated mutations: caffeine-induced $Ca^{2+}$ release, resting

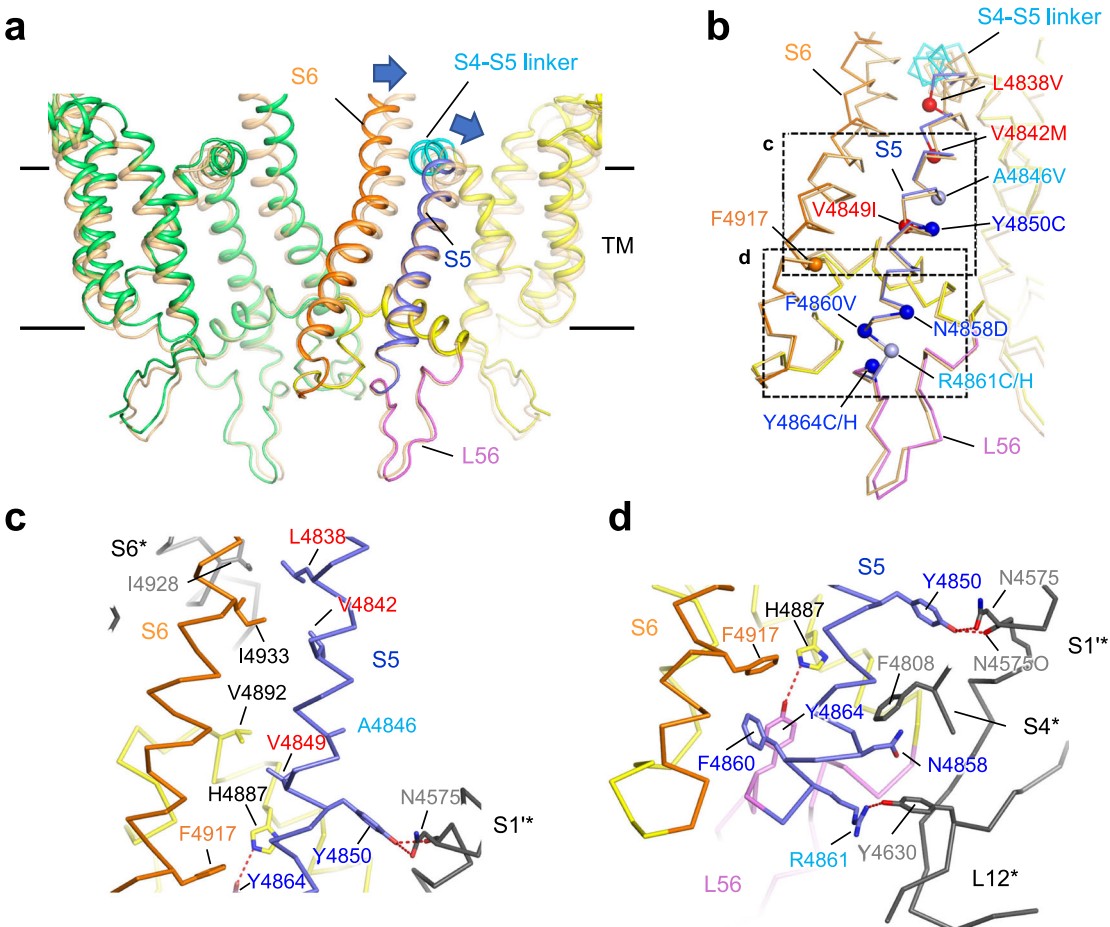

**Fig. 5 | Possible interactions between S5 and the adjacent TM helices based on cryo-EM structures of RyR1. a** Comparison of the closed (PDB accession code, 5TB0) and open (PDB accession code, 5T15) states in the TM region showing two molecules of the tetramer facing each other. The molecule on the left is shown in green for the closed state and brown for the open state. In the molecule on the right, the S4-S5 linker, S5, L56 and S6 in the closed state are shown in light blue, blue, magenta and orange, respectively. Arrows indicate movement during the conformational change from the closed to the open state. **b** Enlarged view from the S4-S5 linker to S6 in the TM region. The color scheme is the same as in (**a**). The kink of S6 occurs after F4917. Blue circles indicate residues with severe loss of function, light blue residues with mild loss of function, and red residues with gain of function. **c** Enlarged view of the region in the closed state enclosed by the square dotted line c in (**b**). Hydrogen bonds are indicated by red dotted lines. **d** Enlarged view of the region in the closed state enclosed by the square dotted line d in (**b**). Hydrogen bonds are indicated by red dotted lines. S1'*, L12*, S4* and S6* in gray are the main and side chains from neighboring molecules.

$[Ca^{2+}]_{cyt}$ and $[Ca^{2+}]_{ER}$ measurements and $Ca^{2+}$-dependent [³H]ryanodine binding (Figs. 2, 3). MH mutants showed enhanced caffeine sensitivity, increased resting $[Ca^{2+}]_{cyt}$, reduced resting $[Ca^{2+}]_{ER}$ and increased [³H] ryanodine binding (Figs. 2 and 3), which are indicative of a gain-of-function phenotype. In contrast, CCD mutants exhibited reduced or no caffeine sensitivity and suppressed [³H]ryanodine binding, indicating a loss-of-function phenotype. Notably, three mutants (Y4850C, F4860V and Y4861H) showed no detectable channel activity (Figs. 2 and 3, Supplementary Table 3). One might speculate that these channels are not properly expressed, folded or localized. However, considering that the expression of these mutants was confirmed by Western blot (Supplementary Fig. 1) and that the CCD mutants in the same or adjacent residues (N4858D and Y4864C) showed small but detectable activity in the caffeine-induced $Ca^{2+}$ release (Fig. 2), we believe that the three mutants may form the functional channel with strongly suppressed activity. Thus, our approach clearly classified the MH and CCD phenotypes.

Loss of DICR is a major cause of CCD[9,39]; therefore, it is essential to determine the DICR activity of mutant channels. However, this assay is difficult because it requires the use of skeletal muscle contexts, such as myotubes from RyR1-deficient (dysgenic) mice[43] or an established cell line (1B5)[44]. We recently developed a reconstituted DICR platform using RyR1-expressing HEK293 cells in which essential components (Cav1.1, β1a, Stac3,

junctophilin) are expressed by baculovirus[21]. Importantly, this platform allows for the quantitative assessment of DICR activity. We showed that CCD mutants inhibit DICR activity to varying degrees depending on the mutation (Fig. 4). Five mutants (Y4850C, N4858D, F4860V, Y4864C and Y4864H) showed no [K⁺]-induced response, indicating a severe loss-of-function mutation. Interestingly, patients carrying the Y4850[28] and N4858D[29] mutations were diagnosed in the neonatal period (2-3 years old) with severe clinical phenotypes. This raises the possibility that DICR activity in CCD may reflect a clinical phenotype. We also observed that MH mutations increase DICR activity (Fig. 4). This is consistent with the previous findings that MH mutations show an enhanced E-C coupling[45,46]. Thus, the reconstituted DICR platform is highly useful for characterizing both gain- and loss-of-function RyR1 mutations.

In the opening of the RyR1 channel, the cytoplasmic side of S5 is aligned and moves coordinately with S6[2,10] (Fig. 5 and Supplementary Movie 1). We demonstrated that three MH mutations (L4838V, V4842M and V4849I) in the cytoplasmic side of S5 showed gain-of-function phenotype. We identified three van der Waals interactions (L4838-I4928, V4842-I4933 and V4849-V4892) with S6 and the pore helix (Fig. 5, Supplementary Fig. 2 and Supplementary Movie 2) and validated that these interactions are involved in gain-of-function of the channel (Fig. 6 and Supplementary Fig. 3). These findings suggest that the cytoplasmic side of S5 interacts with S6 to reduce the

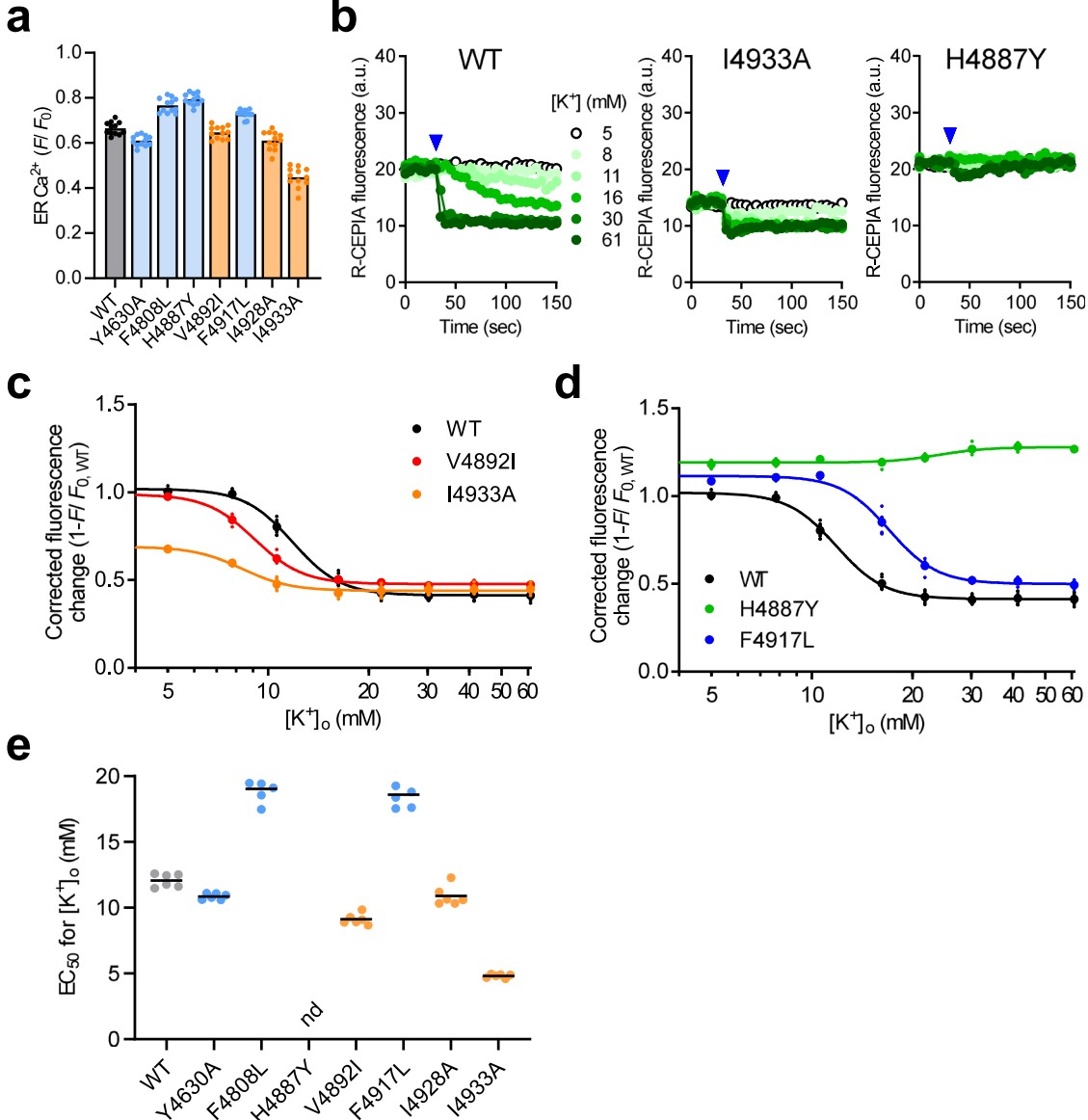

**Fig. 6 | DICR activity of the RyR1 channels carrying mutations in the interacting partners.** **a** $[Ca^{2+}]_{ER}$ of the reconstituted cells expressing WT (black) and mutant RyR1s at the interacting partners for MH (orange: V4892A, I4928A and I4933A) and CCD (light blue; Y4630A, F4808L, H4887Y and F4917L) mutations. **b** Typical results of a time-lapse R-CEPIA1er fluorescence measurement for WT (left), I4933A (center) and H4887Y (right) RyR1 cells. A high $[K^+]$ solution ranging from 5 to 61 mM (symbols shown in WT) was applied at 30 s after starting (blue arrowheads). **c** and **d** $[K^+]$ dependence of R-CEPIA1er fluorescence corrected by $[Ca^{2+}]_{ER}$ in WT and mutant RyR1s (V4892I and I4933A in **c** and H4887Y and F4917L in **d**). **e** $EC_{50}$ values for $[K^+]$. nd, not determined due to virtually no reduction in $[Ca^{2+}]_{ER}$ by $[K^+]$. Data are shown as the means and individual points ($n = 12$, $N = 2$ for (**a**) and $n = 6$, $N = 2$ for (**c**–**e**)). "n" is the number of wells, and "N" is the number of independent experiments.

channel activity. How do the interactions reduce the channel activity? It is possible that the interactions either stabilize the channel in the closed state or destabilize it in the open state, or both. With the current information, we cannot conclude which mechanism is triggered by these mutations. Further structural studies of the mutant channels by cryo-EM would provide clues for this interesting mechanism. Notably, a RyR2 mutation (V4821I) in catecholaminergic polymorphic ventricular tachycardia (CPVT) was identified at the corresponding residue, V4892[47]. CPVT mutations generally result in a gain-of-function phenotype[48]. Thus, role of the cytoplasmic side of S5 may be common to RyR2.

Stable pore formation is essential for ion permeation through the channels. Indeed, CCD mutations in RyR1 within or near the pore loop have been shown to cause severe loss of function[36,49,50]. Five important interactions were identified in the luminal side of S5 (Fig. 5 and Supplementary Table 4). Among them, the Y4864-H4887 interaction may help localize the

pore helix to the appropriate position, consistent with the concept of stable pore formation. In contrast, the remaining interactions (Y4850-N4575, N4858-F4808, F4860-F4917 and R4861-Y4630) are formed with the adjacent transmembrane segments (S1', S4, S6 and S1-S2 loop) of their own and neighboring subunits and do not appear to directly contribute to pore formation. The luminal side of S5 and the aligned luminal side of S6 do not move during channel opening (Fig. 5 and Supplementary Movie 1). The rigid structural base provided by these interactions may be necessary for the displacement of the cytoplasmic side of S6. Given the importance of pore formation and the rigid structural base for channel gating, it is reasonable to assume that mutations in the corresponding interactions lead to severe loss-of-function phenotypes. Interestingly, several CCD or myopathy mutations have also been reported in Y4631, a neighboring residue of the interacting partner (Y4630)[51–53]. Further comprehensive research would provide a complete picture of the mechanism.

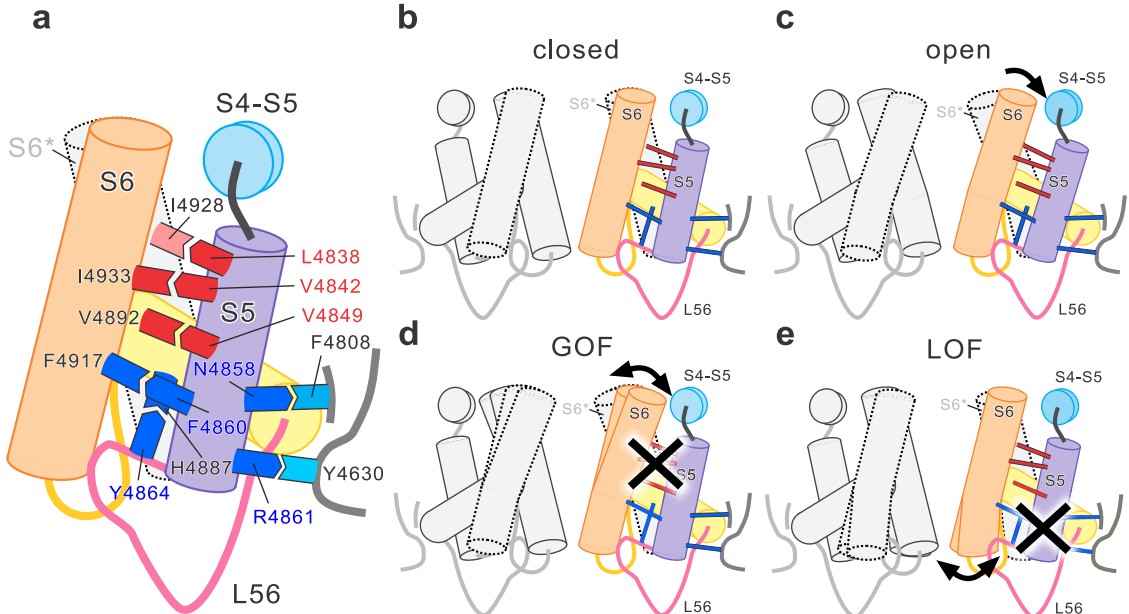

**Fig. 7 | Schematic diagram of the interactions between S5 and the other transmembrane segments. a** All interactions between the amino acid residues studied in this work are shown. The concave and convex rectangles represent amino acid side chains; red and blue represent the gain-of-function (GOF) and loss-of-function (LOF) mutations, respectively. Side chains belonging to neighboring chains (I4928, F4808 and Y4630) are shown in a lighter color. Residues for GOF and LOF mutations in S5 are numbered in red and blue, respectively. Black numbers indicate interacting partners of residues for disease-associated mutations. Each α-helix is shown as a cylinder and loops as lines. **b, c** Schematic diagram in the closed (**b**) and open (**c**) states. Four of the two chains facing each other were shown. Amino acid interactions are represented as straight lines. The meaning of the color of each line is the same as shown in (**a**). **d** Mechanism by which the mutation exhibits GOF; the interaction between the cytoplasmic side in S5 and S6 is altered, resulting in an increased frequency to the open state. **e** Mechanism by which the mutation exhibits LOF; the interaction between the luminal side in S5 and S6 is lost, making the opening of the channel difficult.

We recently identified the fundamental gating mechanism of RyR2, a cardiac isoform[16]. First, the channel activity is reduced by two independent mechanisms: one through interactions in the cytoplasmic domains, that is, U-motif/S6 and U-motif/CTD, and another through transmembrane segment interactions, such as S1/S4 and S3/S4, which regulate the movement of the S4-S5 linker. Second, a series of interactions in the surrounding regions, such as the U-motif/S2-S3 linker domain, S1/S2 and S2/S3, is critical for channel opening in response to $Ca^{2+}$ binding. However, information regarding the role of S5 is lacking. This study successfully added two novel fundamental mechanisms for channel gating through S5. Because the residues involved in important interactions were highly conserved (Supplementary Fig. 4), these mechanisms may be similar to those of other RyR isoforms.

To date, more than 400 disease-associated mutations have been identified in RyR1; however, many remain uncharacterized. Employing a combinatorial strategy that involves functional characterization of mutants coupled with high-resolution structural information has the potential to elucidate the regulatory mechanism of the RyR1 channel. This approach aims to deepen our overall understanding of the gating mechanism of this large channel.

## Materials and Methods
### Generation and maintenance of stable inducible HEK293 cell lines
Stable HEK293 cells expressing WT and mutant RyR1 induced by doxycycline were generated using Flp-In T-REx 293 system (Invitrogen) as described previously[19,20]. Mutations were introduced to rabbit RyR1 cDNA (GenBank accession number X15209.1) by inverse polymerase chain reaction of Nhe I–Cla I fragment (Cs10) or Cla I–EcoRV fragment (Cs11) from the cDNA cassettes encoding the full-length rabbit skeletal muscle RyR1 (pBS-RyR1)[35] using PCR primers listed in Supplementary Table 5. The mutated fragment was then subcloned into an expression vector (pcDNA5/FRT/TO-RyR1). The expression vector was transfected into HEK293 cells, and clones with suitable RyR1 expression were selected and used for the experiments. The cells were maintained in DMEM with 10% fetal calf serum (FCS), supplemented with 100 µg/mL hygromycin and 15 µg/mL blasticidin according to the manufacturer's instructions.

### Western blotting
HEK293 cells were plated on 6-well plates and RyR1 expression was induced the next day with 2 µg/mL doxycycline. After 24 hours, cells were harvested, rinsed with PBS and lysed with Pro-Prep protein extraction solution (iNtRON Biotechnology). After centrifugation at 15,000 rpm for 5 min at 4 °C, the extracted proteins were separated on 3–12% linear gradient polyacrylamide gels and transferred to PVDF membranes. The membranes were probed with the primary antibodies against RyR1 (F-1, Santa Cruz Biotechnology) and calnexin (C4731, Sigma-Aldrich), followed by HRP-conjugated anti-mouse IgG (04-18-18, KPL) and anti-rabbit IgG (074-1516, KPL), respectively. Positive bands were detected by chemiluminescence using ImmunoStar LD (Fujifilm Wako Chemicals) as a substrate.

### Single-cell $Ca^{2+}$ imaging
Single-cell $Ca^{2+}$ imaging was performed in HEK293 cells expressing WT or mutant RyR1 as previously described[19,20,54]. All measurements were performed at 26 °C by perfusing solutions using an in-line solution heater/cooler (Warner Instruments). For measurements of caffeine-induced $Ca^{2+}$ transients, cells loaded with fluo-4 AM were incubated in a HEPES-buffered Krebs solution (140 mM NaCl, 5 mM KCl, 2 mM $CaCl_2$, 1 mM $MgCl_2$, 11 mM glucose and 5 mM HEPES at pH 7.4) and challenged with varied concentrations of caffeine. Fluo-4 was excited at 488 nm through a 20× objective lens, and light emitted at 525 nm was captured with an EM-CCD camera at 700 millisecond intervals (Model 8509; Hamamatsu Photonics, Hamamatsu, Japan). The fluorescence intensity in individual cells ($F$) was determined using ROI analysis with AquaCosmos software (Hamamatsu Photonics). The fluorescence signal ($F$) was normalized to the maximal fluorescence intensity ($F_{max}$) which was obtained with a Krebs solution

containing 20 mM $CaCl_2$ and 20 μM ionomycin at the end of each experiment.

The ER luminal $Ca^{2+}$ concentration ($[Ca^{2+}]_{ER}$) was measured using R-CEPIA1er, a genetically encoded $Ca^{2+}$ indicator[37]. Cells were transfected with R-CEPIA1er cDNA (kindly provided by Dr. Masamitsu Iino) 26–28 h before measurement. Experiments were performed with the same apparatus used for fluo-4 measurements (see above). R-CEPIA1er was excited at 561 nm and light emitted at 620 nm was captured. $F_{min}$ and $F_{max}$ values were obtained with a solution containing 5 mM 1,2-bis(o-aminophenoxy) ethane-N,N,N′,N′-tetraacetic acid (BAPTA) and 20 mM $CaCl_2$, respectively, in the presence of 20 μM ionomycin.

The resting cytoplasmic $Ca^{2+}$ concentration ($[Ca^{2+}]_{cyt}$) was measured using a ratiometric dye, fura-2 AM[19,20,55]. Cells were incubated in a physiological salt solution containing 150 mM NaCl, 4 mM KCl, 2 mM $CaCl_2$, 1 mM $MgCl_2$, 5.6 mM glucose and 10 mM HEPES at pH 7.4. Fluorescence images were captured at 420 nm using an inverted microscope (IX70; Olympus, Japan) equipped with a 40× objective lens (NA 0.95, UPlanSApo/340, Olympus). A cooled charge-coupled device (CCD) camera (Rolera XR, Qimaging, USA) was used to acquire images at a frequency of one frame every 2 seconds. The excitation wavelengths were set to 345 nm and 380 nm. Image analysis was performed using IPLab software (BD Biosciences Bioimaging, Rockville, MD, USA). Individual cells were selected as regions of interest (ROIs), and for each frame, the average fluorescence intensity (F) of each ROI was calculated after subtracting the background intensity. To estimate $[Ca^{2+}]_{cyt}$, we calculated the ratio of F345 to F380 (fluorescence intensity at 345 nm divided by the intensity at 380 nm), as outlined in previous studies[55]. The dissociation constant ($K_D$) for $Ca^{2+}$ was found to be 239 nM, based on in vitro calibration of fura-2 fluorescence[56].

### [³H]Ryanodine binding

[³H]Ryanodine binding was carried out as described[19,20]. Briefly, microsomes isolated from the HEK293 cells were incubated for 5 h at 25 °C with 5 nM [³H]ryanodine in a medium containing 0.17 M NaCl, 20 mM 3-(N-morpholino)-2-hydroxypropanesulfonic acid (MOPSO) at pH 7.0, 2 mM dithiothreitol, 1 mM AMP and various concentrations of free $Ca^{2+}$ buffered with 10 mM ethylene glycol-bis(2-aminoethylether)-N,N,N′,N′-tetraacetic acid (EGTA). Free $Ca^{2+}$ concentrations were calculated using the WEB-MAXC STANDARD (https://somapp.ucdmc.ucdavis.edu/pharmacology/bers/maxchelator/webmaxc/webmaxcS.htm)[57]. The protein-bound [³H] ryanodine was separated by filtering through polyethyleneimine-treated GF/B filters using Micro 96 Cell Harvester (Skatron Instruments). Non-specific binding was determined in the presence of 20 μM unlabeled ryanodine. [³H]Ryanodine-binding data ($B$) were normalized to the maximum number of functional channels ($B_{max}$), which were separately determined by Scatchard plot analysis using various concentrations (3–20 nM) of [³H] ryanodine in a high-salt medium. The resultant $B/B_{max}$ represents the average activity of each mutant.

### DICR assay

The DICR assay was performed as described previously[21]. Briefly, stable HEK293 cells expressing WT or mutant RyR1s were seeded on 96-well clear-bottom black microplates (#3603; Corning) at a density of $3 \times 10^4$ cells/well in 100 μL culture media supplemented with baculovirus solution for R-CEPIA1er. One day after seeding, 100 μL culture media containing 2 μg/mL doxycycline (for induction of RyR1) and baculovirus solutions for Cav1.1 (carrying N617D $Ca^{2+}$-impermeable mutation[58]), β1a, junctophilin-2, Stac3 and Kir2.1 (2 μL each) were added to each well. The MOI was 7.5–15 for each BV strain (depending on the viral titer). After 24 h, the culture media in the wells were replaced with 81 μL of the HEPES-buffered Krebs solution described above, and the microplate was placed in a FlexStation3 fluorometer preincubated at 37 °C. Signals from R-CEPIA1er, which was excited at 560 nm and emitted at 610 nm, were captured every 5 s for 150 s using SoftMax Pro ver. 7. Subsequently, 30 s after starting, 54 μL of the modified Krebs-solution containing different concentrations of high [$K^+$] (5-61 mM at final concentration) or caffeine (0–20 mM at final concentration) was applied to each well. The change in fluorescence was expressed as $F/F_0$ in which averaged fluorescence intensity of the last 25 s (F) was normalized to that of the initial 25 s ($F_0$). [$K^+$] or caffeine dependence was determined by averaging the $F/F_0$ values in 6 wells of each [$K^+$] or caffeine concentration.

### Structural analysis

Structural analysis was performed with the published high-resolution structures of RyR1 in the closed (PDB accession code, 5TB0) and open (PDB accession code, 5T15) states obtained by cryo-EM[14] using COOT[59]. All structure figures and movies were prepared using PyMol 2.5.5 (The PyMOL Molecular Graphics System, http://www.pymol.org).

### Statistics and reproducibility

Sample sizes were described in each figure legend. Data are shown as the means with individual points. One-way analysis of variance (ANOVA), followed by Dunnett's test, was used to compare the groups. Statistical analyses were performed using Prism 9 (GraphPad Software Inc., La Jolla, CA, USA). $P < 0.05$ was considered as statistically significant.

### Data availability

The data that support the findings of this study are available in the Supplementary Data file, and the uncropped blots are provided as Supplementary Fig. 5. All other data are available from the corresponding authors on reasonable request.

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

## Acknowledgements

We thank Ikue Hiraga for the technical assistance. We also thank the Laboratory of Radioisotope Research and the Laboratory of Proteomics and Biomolecular Science, Research Support Center, Juntendo University Graduate School of Medicine, for technical assistance. We would like to thank Editage (www.editage.jp) for English language editing. This study was partly supported by the Japan Society for the Promotion of Sciences KAKENHI (grant numbers 19H03404 and 22H02805 to T.M., 19K07105 and 22K06652 to N.K., 23K24778, 22H05055 and 23K18453 to T.Y. and 21H02411, 22K19375 and 24K02164 to H.Og.); an Intramural Research Grant (2-5 and 5-6) for Neurological and Psychiatric Disorders of NCNP to T.M.; the Vehicle Racing Commemorative Foundation (6114, 6237 and 6303) to T.M and H.Og., and The Naito Foundation to. T.M.

## Author contributions

T.M. and H.Og. conceived and designed the project. T.M., N.K. and T.Y. performed functional analyses. Y.O. and H.Og. analyzed the structural models. H.Oy. provided the RyR1 cDNA cassettes. T.M., Y.O., N.K., T.Y., H.Oy., T.S., and H.Og. interpreted the data. T.M. and H.Og. wrote the manuscript with inputs from all authors. All the authors reviewed the results and approved the final version of the manuscript.

## Competing interests

The authors declare no competing interests.
