## [Peer review file · Communications Biology]

Reviewers' comments:

Reviewer #1 (Remarks to the Author):

The manuscript COMMSBIO-24-0484-T titled “Dual role of the S5 segment in RyR1 channel gating” by Murayama et al. presents a comprehensive investigation into the functional properties of RyR1 mutations associated with malignant hyperthermia (MH) and central core disease (CCD), shedding light on the intricate mechanisms underlying channel gating. Through meticulous experimentation using calcium imaging in HEK293 cells, ryanodine binding measurements, and high-resolution structural analysis, the authors delineate the differential effects of mutations in the S5 segment of RyR1, elucidating its dual role in channel regulation. Their findings not only reveal the distinct phenotypic manifestations of MH and CCD mutations but also provide crucial insights into the intricate interplay between S5 and other transmembrane segments. Additionally, the establishment of a reconstituted system for evaluating depolarization-induced Ca²⁺ release (DICR) further strengthens the study's significance by offering a platform for quantitative assessment of mutant channel activity. By integrating experimental data with structural analyses, the authors not only advance our understanding of RyR1 channel gating but also underscore the potential of their approach in unraveling the pathogenic mechanisms of various RyR isoforms.

Moreover, the manuscript extends its impact by proposing novel fundamental mechanisms governing RyR channel gating through S5, thereby enriching the current understanding of its regulatory role. The identification of critical interactions within S5 and its neighboring segments provides a structural basis for comprehending the intricate dynamics involved in channel opening and stabilization. Furthermore, the authors' exploration of the functional consequences of RyR1 mutations underscores the clinical relevance of their findings, particularly in elucidating the molecular basis of MH and CCD. By bridging functional characterization with structural insights, this study not only contributes significantly to the field of ion channel biology but also highlights the potential of a combinatorial strategy in deciphering the regulatory mechanisms underlying RyR-associated disorders. Overall, the manuscript constitutes a valuable addition to the scientific literature, warranting publication to disseminate its insightful findings and stimulate further research in this critical area of investigation.

However, before the publication, the authors are requested to address the following comments:

-The authors must consider to show the protein expression data for the loss-of-function mutants such as Y4850C, N4858D, F4860V, R4861C, Y4864C, Y4864H, F4808L, F4917L,

I4933A, and H4887Y.

-Please see that in line 111 on page 5 the mutant N4858D is listed as N4848D.

Reviewer #2 (Remarks to the Author):

In the article 'Dual role of the S5 segment in RyR1 channel gating' Murayama T, et al. examined both the MH and CCD mutations of RyR1, corresponding to 'gain-of-function' and 'loss-of-function', located at the S5 segment of the channel. The RyR1 channel is a calcium channel on the sarcoplasmic reticulum membrane. Upon activation, the opening of RyR1 releases Ca²⁺ into the cytoplasm leading to muscle contraction. Mutations of RyR1 are known to cause severe myopathy. These authors discovered that the two different types of mutations are clustered to the 'upper' and 'lower' parts of S5, and mutations around these two clusters emulate their defects. The overall experiments possibly support the authors' hypothesis, while interpretations fall short.

Major concerns

1. Be very mindful that the structural data are not the original work in this manuscript. Having 2 out of 6 figures completely re-drawn from earlier work can be misleading.
2. As mentioned by the authors MH mutations respond very strongly to agonists and release a huge amount of Ca²⁺ into cytoplasm. However, V4842M and V4849I, studied in this work behave mildly more active than the wild-type channel. These mutants also respond better to depolarization than agonists. What is the explanation for the discrepancy? To me, these two mutants seem to be leaky with reduced Ca²⁺ release rather than being super active. Should they be characterized as gain-of function or loss-of-function? Please clarify.
3. The interpretations of structural data fall short. For example, to make a channel more active a mutation can destabilize the inactive state or stabilize the active state or both. The authors concluded MH mutations destabilized the inactive state without discussing other possibilities. Another example, to make a channel less active mutants may not express well, not fold properly, not be delivered to the right locations, or simply destabilize the active state... The authors did not test or discuss any of these possibilities, especially some of CCD mutants do not show activities at all.
4. The studied interactions were only examined in the inactive (closed) structure of the channel. Wouldn't it make more sense to look at the active (open) state? Please show these interactions listed in Fig.5c and Fig5d in the active structure.
5. Line151, V4842M and V4849I mutations disrupt 'van der Waals interactions'. This is a very exaggerated statement. Both the Valine to Methionine mutation and the Valine to Isoleucine mutation are usually considered minor changes of van der Waals interaction, because all of them are classified as hydrophobic residues with limited side chain lengths. In common sense, these mutations would be predicted to cause minor change

of channel activities, which is evidently true based on the data of this work. Again, please refer to comment 2, these two mutations are ill-explained in this work.

Minor comments

1. It is better to articulate the 'cytoplasmic' or 'sarcoplasmic' side instead of 'upper' or 'lower' part of the S5 segment.
2. In Fig2f, the authors switched to use Fura-2 to report the cytoplasmic Ca²⁺ concentration, but the results should have been already obtained from Fig 2d using Fluo-4 dye. Why, and what are the results from using Fluo-4?

Reviewer #3 (Remarks to the Author):

The manuscript "Dual role of the S5 segment in RyR1 channel gating" is an interesting and important work that describes the effect of the missense mutations in the type 1 Ryanodine receptor linked to severe muscle diseases. The authors combine structural and functional experiments using different calcium dyes to study the effect of the mutations not only in the cytoplasmic calcium but also in the subcellular regions (i.e. endoplasmic reticulum)

Some parts of the experimental protocols is sometimes lacking the necessary details, follow the statement I have some general comments:

- Structural analyses:

-It is not clear which atomic structure has been used for the analyses in Fig 1, 5 and for the supplementary materials, please provide the identification code (e.g. PDB number). Was it a model based on atomic structure or was it a published structure? Which conformational state was the original structure at? Please clarify.

-Despite showing Fig 5 and various supplementary movies of the structure, and various structural analyses (e.g. the identification of amino acids interactions e.g. van der Waals) the authors do not explain how these experiments were undertaken. There is no structural analyses section in the method, did the authors perform molecular dynamics modelling for the movies? Or were these movies from other published sources?

- Cell culture:

-The authors explain how they generated stable cell lines, however they do not explained how they were maintained (e.g. which media etc), please clarify.

- Data analyses:

-In the data analyses section the authors mentioned they used parametric test, either the unpaired Student's t-test or ANOVA followed by the Dunnett's test. Both tests

assume that the data are normality distributed, did the authors verify this assumption? If the data are not normally distributed the non-parametric tests should be employed. -In Fig 4, were different wells used for the [K+] concentration response curve? Also, which “n” (number of wells or N independent experiments) was used for statistical analyses?

Minor comments:

- In the first paragraph of the results the authors write “we identified 12 disease causing mutations”, could the authors please clarify how these mutations were defined as disease-causing? Were they associated in patients with the disease of interest or were they predicted to be pathogenic through bioinformatics tools (e.g. Polyphen 2 etc?)
- It is not clear which concentration of caffeine is used in Figure 1d, please clarify.
- In some figures it is specified how many biological replicates were done (e.g. Fig 4) in others it is not written. For clarity, if possible, could the authors add the number of independent experiments undertaken across the distinct figures?

Responses to reviewers

We sincerely appreciate the reviewers for their insightful and constructive comments. We have revised our manuscript according to the reviewers' comments. Changes in the manuscript are marked in yellow.

In addition, we have added new data and a figure as follows.

1. We added a set of data for new MH mutant (L4838V) (**Figs. 2-4, Supplementary Tables 1-2**), in which we found in the database after submitting the original manuscript and a mutant at the interacting partner (I4928A) (**Fig. 6 and Supplementary Fig. 3, Supplementary Table 3**). This strengthens our conclusion of dual role of the S5 segment.
2. We have added schematic drawings as a new figure (**Fig. 7**). This would help to understand dual role of S5 segment in the channel gating.

Reviewer #1 (Remarks to the Author):

The manuscript COMMSBIO-24-0484-T titled “Dual role of the S5 segment in RyR1 channel gating” by Murayama et al. presents a comprehensive investigation into the functional properties of RyR1 mutations associated with malignant hyperthermia (MH) and central core disease (CCD), shedding light on the intricate mechanisms underlying channel gating. Through meticulous experimentation using calcium imaging in HEK293 cells, ryanodine binding measurements, and high-resolution structural analysis, the authors delineate the differential effects of mutations in the S5 segment of RyR1, elucidating its dual role in channel regulation. Their findings not only reveal the distinct phenotypic manifestations of MH and CCD mutations but also provide crucial insights into the intricate interplay between S5 and other transmembrane segments. Additionally, the establishment of a reconstituted system for evaluating depolarization-induced Ca²⁺ release (DICR) further strengthens the study's significance by offering a platform for quantitative assessment of mutant channel activity. By integrating experimental data with structural analyses, the authors not only advance our understanding of RyR1 channel gating but also underscore the potential of their approach in unraveling the pathogenic mechanisms of various RyR isoforms.

Moreover, the manuscript extends its impact by proposing novel fundamental mechanisms governing RyR channel gating through S5, thereby enriching the current understanding of its regulatory role. The identification of critical interactions within S5 and its neighboring segments provides a structural basis for comprehending the intricate dynamics involved in channel opening and stabilization. Furthermore, the authors' exploration of the functional consequences of RyR1 mutations underscores the clinical relevance of their findings, particularly in elucidating the molecular basis of MH and CCD. By bridging functional characterization with structural insights, this study not only contributes significantly to the field of ion channel biology but also highlights the potential of a combinatorial strategy in deciphering the regulatory mechanisms underlying RyR-associated disorders. Overall, the manuscript constitutes a valuable addition to the scientific literature, warranting publication to disseminate its insightful findings and stimulate further research in this critical area of investigation.

However, before the publication, the authors are requested to address the following comments:

-The authors must consider to show the protein expression data for the loss-of-function mutants such as Y4850C, N4858D, F4860V, R4861C, Y4864C, Y4864H, F4808L, F4917L, I4933A, and H4887Y.

Thank you very much for your suggestions. We performed additional Western blotting experiments on the mutants used in this study and confirmed that all the mutants are expressed (**Supplementary Fig. 1**) (lines 80-81, 189-190 and 310-319).

-Please see that in line 111 on page 5 the mutant N4858D is listed as N4848D.

Thank you for your pointing out our error. We fixed it (line 117).

Reviewer #2 (Remarks to the Author):

In the article 'Dual role of the S5 segment in RyR1 channel gating' Murayama T, et al. examined both the MH and CCD mutations of RyR1, corresponding to 'gain-of-function' and 'loss-of-function', located at the S5 segment of the channel. The RyR1 channel is a calcium channel on the sarcoplasmic reticulum membrane. Upon activation, the opening of RyR1

releases Ca²⁺ into the cytoplasm leading to muscle contraction. Mutations of RyR1 are known to cause severe myopathy. These authors discovered that the two different types of mutations are clustered to the 'upper' and 'lower' parts of S5, and mutations around these two clusters emulate their defects. The overall experiments possibly support the authors' hypothesis, while interpretations fall short.

Major concerns

1. Be very mindful that the structural data are not the original work in this manuscript. Having 2 out of 6 figures completely re-drawn from earlier work can be misleading.

Thank you for raising this concern. As you point out, the structural data used in this paper were based on the published structures. We cited the PDB accession codes (5TB0 and 5T15 for closed and open states, respectively) in Results (lines 157-158), Figure legends (lines 386 and 440-441), Supplementary figure legends (line 490) and Supplementary movie legends (lines for 508-509 and 512). We also added clear statements in the abstract that the experiments in this paper were interpreted with respect to the published RyR1 structure (lines 20-21).

2. As mentioned by the authors MH mutations respond very strongly to agonists and release a huge amount of Ca²⁺ into cytoplasm. However, V4842M and V4849I, studied in this work behave mildly more active than the wild-type channel. These mutants also respond better to depolarization than agonists. What is the explanation for the discrepancy? To me, these two mutants seem to be leaky with reduced Ca²⁺ release rather than being super active. Should they be characterized as gain-of-function or loss-of-function? Please clarify.

We added a statement that caffeine sensitivity is generally enhanced in MH mutants (lines 86-88). We added a typical trace and dose-dependent plot for V4842M (**Fig. 2b, c**), which clearly shows enhanced caffeine sensitivity (lines 89-90). The three MH mutants (we analyzed another MH mutant L4838V in addition to V4842M and V4849I during revision) showed enhanced sensitivity to caffeine (**Fig. 2c**) (lines 97-98). In addition, we newly analyzed EC₅₀ and IC₅₀ values for Ca²⁺ in [³H]ryanodine binding and found that EC₅₀ was significantly reduced with two MH mutants (L4838V and V4849I) (**Fig. 3d**), whereas IC₅₀ was increased with the three MH mutants (**Fig. 3e**) (lines 118-121). Finally, we calculated the relative activity of the mutant channels at physiological [Ca²⁺]_{cyt} (pCa 7) and found that L4838V, V4842M and V4849I, showed 10-, 2-, and 10-fold higher activity than WT, respectively (**Fig.**

3f) (lines 121-125). Taken together, these results suggest that these mutants are characterized by mild (V4842M) and severe (L4838V and V4849I) gain-of-function mutations.

3. The interpretations of structural data fall short. For example, to make a channel more active a mutation can destabilize the inactive state or stabilize the active state or both. The authors concluded MH mutations destabilized the inactive state without discussing other possibilities. Another example, to make a channel less active mutants may not express well, not fold properly, not be delivered to the right locations, or simply destabilize the active state... The authors did not test or discuss any of these possibilities, especially some of CCD mutants do not show activities at all.

Thank you very much for your insightful suggestion. In the original manuscript, we only interpreted that interactions affected by MH mutations (V4842M and V4849I) stabilize the channel in the closed state. However, as you suggested, it is also possible that these interactions could destabilize the channel in the open state. We described both possibilities in Result (lines 168-169) and Discussion (lines 253-258). We also changed the relevant descriptions in the Abstract (lines 22-23), Introduction (lines 65-66), and Discussion (lines 214-215). Also, we believe that the addition of new experiments on the MH mutation (L4838V) strengthens our statement.

For three CCD mutants (Y4850C, F4860V and Y4861H) which showed any detectable channel activity, it is possible that the channels are not properly folded or localized. However, considering that the CCD mutants in the same or neighboring residues (N4858D and Y4864C) showed small but detectable activity in the caffeine-induced Ca^{2+} release (**Fig. 2**), and Western blot analysis exhibited that the expression levels of these mutants were similar to that of WT (**Supplementary Fig. 1**), we believe that the three mutants may form the functional channel with strongly suppressed activity. We discussed this in Discussion (lines 222-229).

4. The studied interactions were only examined in the inactive (closed) structure of the channel. Wouldn't it make more sense to look at the active (open) state? Please show these interactions listed in Fig.5c and Fig5d in the active structure.

Thank you for your constructive feedback. We have now created **Supplementary Fig. 2**, which corresponds to **Fig. 5c** and **Fig. 5d** in the open state. This figure shows that the important interactions revealed in the closed form are also preserved in the open form. This is

a good indication that this part is structurally important. We described this in Results (lines 166, 174-175).

5. Line151, V4842M and V4849I mutations disrupt ‘van der Waals interactions’. This is a very exaggerated statement. Both the Valine to Methionine mutation and the Valine to Isoleucine mutation are usually considered minor changes of van der Waals interaction, because all of them are classified as hydrophobic residues with limited side chain lengths. In common sense, these mutations would be predicted to cause minor change of channel activities, which is evidently true based on the data of this work. Again, please refer to comment 2, these two mutations are ill-explained in this work.

Thank you for your pointing out. We agree with you that both the Valine to Methionine mutation and the Valine to Isoleucine mutation are considered minor changes of van der Waals interaction. We therefore changed the word “disrupt” to “may weaken or disrupt” (lines 167, 178 and 185). However, as stated in an answer to major concern #2, these mutations cause mild (V4842M) or severe (V4849I) gain-of-function. Moreover, the mutations in the counter residues (V4933A, V4892I) also yield gain-of-function phenotype. Based on these findings, we believe that van der Waals interactions in these residues are important in the channel gating. Similar results of an additional MH mutant, L4838V and its counterpart, I4928A strengthen our conclusion.

Minor comments

1. It is better to articulate the ‘cytoplasmic’ or ‘sarcoplasmic’ side instead of ‘upper’ or ‘lower’ part of the S5 segment.

Thank you very much for your suggestion. We changed cytoplasmic and luminal side instead of upper and lower part of the S5 segment (lines 22 and 23 at first appearance).

2. In Fig2f, the authors switched to use Fura-2 to report the cytoplasmic Ca²⁺ concentration, but the results should have been already obtained from Fig 2d using Fluo-4 dye. Why, and what are the results from using Fluo-4?

Fluo-4 was used in caffeine-induced Ca²⁺ release experiments and **Fig. 2d** shows the released Ca²⁺ by caffeine. On the other hand, we measured “resting” cytoplasmic Ca²⁺ using fura-2 in **Fig. 2g** (now we exchanged **Fig. 2f** and **Fig. 2g**), which is an index of Ca²⁺ influx by ER Ca²⁺

depletion. To clarify these measurements, we described additional explanations in Results (lines 105-107) and Methods (line 334).

Reviewer #3 (Remarks to the Author):

The manuscript “Dual role of the S5 segment in RyR1 channel gating” is an interesting and important work that describes the effect of the missense mutations in the type 1 Ryanodine receptor linked to severe muscle diseases. The authors combine structural and functional experiments using different calcium dyes to study the effect of the mutations not only in the cytoplasmic calcium but also in the subcellular regions (i.e. endoplasmic reticulum) Some parts of the experimental protocols is sometimes lacking the necessary details, follow the statement I have some general comments:

· Structural analyses:

-It is not clear which atomic structure has been used for the analyses in Fig 1, 5 and for the supplementary materials, please provide the identification code (e.g. PDB number).

We thank you for your kind suggestion. We added the respective PDB accession codes in the legends for **Fig. 1**, **Fig. 5**, and **Supplementary Fig. 2**. We have also indicated them in Results (lines 157-158).

Was it a model based on atomic structure or was it a published structure? Which conformational state was the original structure at? Please clarify.

We thank you for your constructive suggestion. All structures used for our interpretations of mutant experiments in this paper are published structures. We added clear statements in Abstract (line 20-21) and Results (line 157-158) that the experiments in this paper were interpreted with respect to the published RyR1 structure.

-Despite showing Fig 5 and various supplementary movies of the structure, and various structural analyses (e.g. the identification of amino acids interactions e.g. van der Waals) the authors do not explain how these experiments were undertaken. There is no structural analyses section in the method, did the authors perform molecular dynamics modelling for the movies? Or were these movies from other published sources?

Thank you for your pointing out. We have added to the Methods section the software used to analyze the structure and the software used to prepare the figures and movies (lines 370-374).

· Cell culture:

-The authors explain how they generated stable cell lines, however they are not explained how they were maintained (e.g. which media etc), please clarify.

We added the description of maintenance of the cells in Methods (lines 297 and 307-309).

· Data analyses:

-In the data analyses section the authors mentioned they used parametric test, either the unpaired Student's t-test or ANOVA followed by the Dunnett's test. Both tests assume that the data are normality distributed, did the authors verify this assumption? If the data are not normally distributed the non-parametric tests should be employed.

Thank you for your suggestions. In the case of Ca²⁺ imaging (**Fig. 2**), where number of data are large, we know empirically that the data are considered normally distributed. Therefore, we analyzed them by ANOVA followed by the Dunnett's test. For experiments with small n numbers, i.e., [³H]ryanodine binding (**Fig. 3**) and DICR assay (**Figs, 4, 6 and Supplementary Fig. 3**), we re-analyzed the data by non-parametric Mann-Whitney test (lines 377-381).

-In Fig 4, were different wells used for the [K+] concentration response curve? Also, which "n" (number of wells or N independent experiments) was used for statistical analyses?

We used different wells for the [K+] concentration response curve. "n" number was used for statistical analysis (lines 368-369).

Minor comments:

· In the first paragraph of the results the authors write "we identified 12 disease causing mutations", could the authors please clarify how these mutations were defined as disease-causing? Were they associated in patients with the disease of interest or were they predicted to be pathogenic through bioinformatics tools (e.g. Polyphen 2 etc?)

These mutations have been identified in the patients but not fully defined as disease-causing. We changed the term from “disease-causing” to “disease-associated” (line 20 at first appearance).

· It is not clear which concentration of caffeine is used in Figure 1d, please clarify.

Thank you for your pointing out. We labeled concentrations of caffeine (10 mM) in the figure.

· In some figures it is specified how many biological replicates were done (e.g. Fig 4) in others it is not written. For clarity, if possible, could the authors add the number of independent experiments undertaken across the distinct figures?

We added the “N” number in each experiment. See figure legends in Figs. 2 and 3.

REVIEWERS' COMMENTS:

Reviewer #2 (Remarks to the Author):

The authors' rebuttal carefully addressed my previous comments with additional data, figure and clarifications.

Regarding the last major comment that I had previously, I would like to re-iterate that V4842M and V4849I mutations disrupt 'van der Waals interactions' is an exaggerated statement to readers who have chemistry or structural biology background. I can't tell from the new submission how the authors calculated free energy to use "weaken" instead. More appropriate wording includes 'alter, change, differ..' This is consistent with another argument of mine; based on the available data it is impossible to tell whether mutations affect the closed state or the active state of the channel.

Reviewer #3 (Remarks to the Author):

The authors addressed my comments

Response to the reviewer

Reviewer #2 (Remarks to the Author):

Regarding the last major comment that I had previously, I would like to re-iterate that V4842M and V4849I mutations disrupt 'van der Waals interactions' is an exaggerated statement to readers who have chemistry or structural biology background. I can't tell from the new submission how the authors calculated free energy to use "weaken" instead. More appropriate wording includes 'alter, change, differ...' This is consistent with another argument of mine; based on the available data it is impossible to tell whether mutations affect the closed state or the active state of the channel.

Thank you for your kind explanation. According to your suggestion, we changed the word to “alter” (line 170).